# Long-Term Management of Post-Stroke Spasticity with Botulinum Toxin: A Retrospective Study

**DOI:** 10.3390/toxins16090383

**Published:** 2024-09-03

**Authors:** Nicoletta Falcone, Fabrizio Leo, Carmelo Chisari, Stefania Dalise

**Affiliations:** 1Department of Traslational Research and New Technologies in Medicine and Surgery, Unit of Neurorehabilitation, University of Pisa, 56126 Pisa, Italy; carmelo.chisari@unipi.it; 2Neurorehabilitation Unit, Department of Neuroscience, University Hospital of Pisa, 56124 Pisa, Italy; fabrizio.leo@gmail.com

**Keywords:** botulinum toxin A, long-term management, treating, stroke, spasticity

## Abstract

Stroke-induced spasticity is a prevalent condition affecting stroke survivors, significantly impacting their quality of life. Botulinum Toxin A injections are widely used for its management, yet the long-term effects and optimal management strategies remain uncertain. This retrospective study analyzed medical records of 95 chronic stroke patients undergoing long-term BoNT-A treatment for spasticity. Demographic data, treatment duration, dosage variability, and dropout rates were assessed over a period ranging from 2 to 14 years. The study revealed a notable extension of the interval between BoNT-A injections throughout the treatment duration. Dropout rates peaked during the initial 5 years of treatment, perhaps due to perceived treatment ineffectiveness. Additionally, a trend of escalating dosage was observed across all groups, indicating a potential rise in the severity of spasticity or changes in treatment response over time. BoNT-A injections emerged as the predominant treatment choice for managing post-stroke spasticity. The delayed initiation of BoNT-A treatment underscores the need for heightened awareness among healthcare providers to recognize and manage spasticity promptly post-stroke. Patients’ expectations and treatment goals should be clearly defined to optimize treatment adherence, while the observed escalation in dosage and treatment intervals emphasizes the dynamic nature of spasticity and underscores the importance of monitoring long-term treatment outcomes.

## 1. Introduction

Stroke is the second leading cause of both disability and death worldwide, with the highest burden of the disease shared by low- and middle-income countries [1].

Post-stroke spasticity (PSS) is a frequent clinical sign in stroke survivors, but estimates of incidence and prevalence vary widely. This is perhaps due to differences in the definition and clinical measurement of spasticity [2].

There is evidence of PSS in 4–27% of patients in the acute phase (1–4 weeks after the stroke), 19–26.7% of those in the subacute phase (1–3 months after stroke), and 17–42.6% of those in the chronic phase (>3 months after stroke) [3].

In the 1980s, spasticity was defined by J. W. Lance as a “motor disorder characterized by a velocity-dependent increase in tonic stretch reflexes as one component of the upper motor neuron syndrome” [4].

However, this definition has failed to comprehensively explain the complexity of the spasticity phenomenon, prompting some authors to propose new and more detailed definitions. As stated by Dressler et al., involuntary muscle hyperactivity may encompass spasticity sensu strictu, rigidity, dystonia, or spasms [5]. These phenomena significantly affect the individual’s functionality and quality of life [6] and may limit the potential success of rehabilitation [7].

Regardless of the definition given, in a clinical setting, it is important to understand whether spasticity is more beneficial or deleterious and to establish treatment in accordance with the principles of a patient-centered goal setting [8].

There are currently several treatment options for spasticity, including muscle relaxant drugs (e.g., Baclofen or Tizanidine); passive static/dynamic stretching; the Botulinum Toxin A injection; Extracorporeal Shock Wave Therapy (ESWT); Non-Invasive Brain Stimulation (NIBS); Localized Muscle Vibration (LMV), and when conservative therapies are no longer effective, for example due to tendon retraction, a surgical approach should be considered [9].

Botulinum Neurotoxin-A (BoNT-A) is considered the gold standard treatment for focal spasticity since it overcomes some of the limits of the aforementioned alternative therapies. For instance, compared to ESWT [10,11] and NIBS, its mechanism is fully understood, whereas we still require more knowledge about the optimal treatment settings of ESWT, as well as some confirmation on the duration efficacy of NIBS [9]. Similarly, more research is needed regarding the optimal treatment settings and the duration of the effects of the LMV. Finally, muscle relaxant drugs present relevant non-focal adverse effects [9]. BoNT-A, on the contrary, can produce focal, controllable muscle weakness of predictable duration with minimal (mainly local) adverse effects [12].

For these reasons, its use is supported by high-quality GRADE A evidence in both the upper and lower limbs [9,13].

BoNT-A injections are commonly used to manage focal spasticity after a stroke. The commercially available neurotoxins include three main brands of BoNT/A: OnabotulinumtoxinA (BOTOX^®^), AbobotulinumtoxinA (DYSPORT^®^), and IncobotulinumtoxinA (XEOMIN^®^) [14].

These preparations contain different excipients but have similar molecular architecture [15] and mechanisms of action, which inhibit the release of acetylcholine in neuro-muscular junctions [16]. Their potency may vary based on the amount of active neurotoxin and the presence of complex proteins [17].

OnabotulinumtoxinA and IncobotulinumtoxinA have comparable efficacy with a 1:1 conversion ratio. The conversion ratio between Abobotulinum Toxin-A and OnabotulinumtoxinA is still debatable, but a ratio of ≤1:3 seems most appropriate [16].

The efficacy and safety of BoNT-A were extensively investigated in previous studies. To date, a well-established body of evidence demonstrates that BoNT-A is effective in reducing spasticity in both the upper and lower limbs, and that the treatment is safe and generally well tolerated even when a high dosage is required [18,19,20,21,22,23].

Botulinum Toxin injections could also promote functional improvement [19,24,25], but in this regard, there is still conflicting data and limited evidence [26,27].

Studies on the effects of the toxin have traditionally focused on short-term effects per se or its effects in conjunction with rehabilitative treatments, with an average follow-up of around 8 weeks, while the long-term use of the toxin is less investigated.

To the best of our knowledge, only three studies have reported data on the treatment of PSS, with BoNT-A focusing on efficacy over a relatively long-term period ranging from 2 to 9 years, either alone or in combination with physical therapy [28,29,30]. Longer observation periods are available only in studies that include the spasticity of different etiologies [31].

While there is compelling evidence of BoNT-A efficacy in reducing spasticity, further data are required to understand its long-term effects. Specifically, investigating dosage variations and injection intervals over extended therapy periods could provide insights into the duration of effective treatment and any changes in patient response over time, including factors such as muscle modifications.

In this study, we aim to retrospectively analyze our experience with long-term Botulinum Neurotoxin-A treatment in patients with chronic post-stroke spasticity. Our focus is on observing potential changes in BoNT-A treatment over time, particularly in terms of dosage variability and administration interval. This information is valuable for optimizing spasticity management and informing long-term treatment strategies.

## 2. Results

### 2.1. Patient Demographics and Dropout Data

Table 1 summarizes the demographic and treatment-associated data of our sample participants.

Table 2 and Figure 1 show the dropout information over the time course of the treatment. Almost 50% of the dropouts occurred in the first 4 years of treatment (SIT group). This percentage rises to 73% if we consider the first 5 years of treatment. It is worth noting that 88.9% of the dropouts due to ineffective treatment occurred in this time frame. Another cause of dropout was the death of patients (10.8%), and unfortunately, in 64.9% of cases, it was not possible to identify the reason for the dropout.

Appendix A shows the number of patients undergoing treatment with the different brands of BoNT/A in each group.

### 2.2. Toxin Treatment in the Long-Term Course

For each year of treatment (from 1 to 14), we computed the mean interval between injections (see Figure A1 in Appendix B) and the mean administered dosage of the toxin (Figure A2 in Appendix B). We then assessed the temporal evolution of these treatment variables for the whole cohort of patients as well as separately for the SIT and LIT groups.

Figure 2 shows how the interval between injections varied in the first and in the last year of treatment. When considering all the patients, the interval between injections increased at the end of the treatment (5 {1.8}) compared to the beginning (4 {2.45}) (V = 1091, *p* = 0.0008; see Figure 2A). When stratifying the patients in groups based on the duration of the treatment (see Figure 2B), we still observed an increment in the interval of the SIT group (first year: 4 {2.17}, last year: 5.08 {2}, V = 257.5, pfdr = 0.018) and the LIT group (first year: 4.5 {2.34}, last year: 5 {1.92}, V = 292.5, *p* = 0.029).

In order to directly compare the effect of time between groups, we calculated the difference between the interval in the last and in the first year of treatment for each group (delta) (SIT: 0.76 {2.36}, LIT: 0.98 {2.51}). The Mann–Whitney test showed that the two groups did not differ in delta (*p* = 0.98).

Figure 3 shows how the last dose of the treatment varied compared to the first. When considering all the patients, the dosage increased at the end of the treatment (500 {700}) compared to the beginning (500 {800}) (V = 800, *p* = 0.002; see Figure 3A). When stratifying the patients in groups based on the duration of the treatment (see Figure 3B), we still observed an increment in the dosage for the SIT group (first dose: 500 {800}, last dose: 500 {700}, V = 188.5, pfdr = 0.046) and a trend towards the same increment in the LIT group (first dose: 500 {750}, last dose: 500 {700}, V = 227.5, *p* = 0.06). This result indicates that even shorter durations of treatment (such as those of the SIT group) already require an increment in the dosage, which then tends to stabilize for longer durations.

In order to directly compare the effect of time between groups, we computed the variation in the last dose compared to the first expressed as a percentage. This normalization was due to the fact that the baseline dosage of the three different toxins we used was not equal. There was, indeed, a 3:1 ratio between the dosage of the AbobotulinumtoxinA and the dosage of the other two toxins. Since our groups had slightly different ratios of number of patients using the different toxins (see Appendix A), we had to normalize the variations using percentages. The Mann–Whitney test showed that the two groups did not differ in terms of variation between the first and last dosage (SIT: 20 {61.7}, LIT: 0 {68.6}) (W = 1021, *p* = 0.42).

Finally, a small number of patients (7 out of 95) also took anti-spastic drugs (Baclofen, Tizanidine, or Eperisone Hydrochloride) during their treatment with BoNT-A. Importantly, this subgroup was almost equally distributed across the SIT (n = 3) and LIT (n = 4) groups, which minimized the risk of bias. The small sample size prevented us from making statistical comparisons, and we only performed some descriptive analyses. The duration of the treatment of the group who used anti-spasticity drugs was 6.7 ± 3.8, whereas the duration of the treatment of the group who did not use such drugs was 5.2 ± 3.1. Figure A3 and Figure A4 in Appendix B show the intervals and dosages of individual patients who took or did not take anti-spasticity drugs, respectively.

## 3. Discussion

The purpose of our retrospective study was to help fill the current lack of information regarding the role of Botulinum Toxin injections in the long-term management of spasticity in chronic post-stroke patients. Therefore, we performed a review of the records of patients with chronic stroke who were treated with repeated injections of BoNT-A for a minimum period of 2 years and a maximum of 14 years.

From our analysis, it emerged that in the first 5 years of treatment, the dropout percentage was significantly higher than in subsequent years. We could hypothesize that this phenomenon is due to the fact that, in the first years of the disease, many patients have not yet completely accepted the functional outcomes and have very high expectations regarding the BoNT-A treatment; when these expectations are not met, patients’ motivation weakens, leading to an abandonment of treatment. Supporting this hypothesis is the fact that about 89% of dropouts due to the ineffectiveness of the treatment happened during the first 5 years.

As suggested by Levy et al., patients hampered by disabling spasticity but whose underlying motor control remained satisfactory may expect a realistic functional recovery after BoNT-A injection [27]; in patients with poor residual motor control, however, the objectives achievable with treatment are represented by the prevention of pathological postures and skin sores, as well as the facilitation of hygiene and dressing maneuvers [26].

Therefore, to maintain good compliance, it is essential to define clear objectives before starting BoNT-A treatment and to ensure that the patient shares them.

Furthermore, our results show that in the entire group, the average interval between treatments increased significantly in the last year compared to the first.

This result is in line with Lagalla et al. in which the effects of repeated injections of BoNT-A were analyzed over time with a 3-year follow-up [29].

The increase in the interval time between injections in patients who continue treatment cannot be interpreted in a straightforward manner. Although follow-up and subsequent treatments are generally planned approximately 3–4 months after the previous one, various factors can influence the timing. These factors include the progressive general worsening of the patient’s health due to aging and the onset of further comorbidities, which may make the treatment of spasticity a lower priority, consequently leading to postponed appointments. Furthermore, the extension of intervals between treatments could be due to patients losing interest in a therapy that does not sufficiently relieve their symptoms.

Indeed, the toxin’s effect may be most pronounced initially, most likely due to the muscle’s initial integrity, which can lead to greater patient adherence to treatment; conversely, over a longer observation period, changes in the intrinsic properties of the muscle, a concept that will be further elucidated later in the text, may reduce responses to treatment and result in shorter treatment intervals. This concept might seem inconsistent with our results, but since a reduced response may potentially lead to a decrease in a patient’s adherence to treatment, it can explain the increased intervals between injections.

The most interesting finding from our study is the fact that during long-term treatment, there was a gradual and significant increase in the dosage of Botulinum Toxin administered to patients. This increase is highlighted both by observing the group of patients as a whole and by considering the two groups separately, suggesting that even relatively short treatment durations (around 4 years) are sufficient to observe an increase in dosage.

In particular, when considering the two groups separately, the SIT group showed a significant increment in the dosage, while in the LIT group, we only observed a trend towards the same increment. This could be influenced by the fact that during the first treatment, there is a tendency to start from low doses and then titrate the BoNT-A dosage during subsequent treatments, making the increase in dosage in the SIT group more pronounced than in the LIT group.

The general increase in dosage over time, however, could be explained by the worsening of spasticity, which gradually involves an ever-increasing number of muscles. In a previous study, it was observed that spasticity is a dynamic phenomenon and that its degree of severity often changes over time in both directions [32]. However, the decrease or normalization of tone is not common and occurs predominantly in patients with mild spasticity, while in patients who have moderate spasticity from the early stages of the disease, this symptom is more likely to worsen over time [33].

Furthermore, it should be pointed out that in [32,33], the follow-up was 3 and 12 months, respectively, while our data refer to a much longer time span. Therefore, it cannot be ruled out that this symptom could be more likely to worsen in the long-term due, in part, to maladaptive plasticity phenomena [34].

On the other hand, the need to increase the dosage could also indicate a progressively reduced response to the treatment, which could also depend on the structural changes that occur in the spastic muscle over time.

The muscles of patients with chronic stroke present morphological alterations (sarcopenia and rigidity), metabolic alterations (the increased tissue production of lactate and glycerol, delayed and reduced glucose utilization, a shift towards a low content of type IIX oxidative fibers), and electromechanical changes (changes in motor unit activation). Nevertheless, to date, data on this topic still show some conflicting findings [35].

Furthermore, studies on animal models have revealed that long-term treatment with Botulinum Toxin type A induces changes in the structure and mechanics of both target and non-target muscles (e.g., increased intramuscular collagen and muscle atrophy) [36].

Nonetheless, the role of BoNT-A treatment on human muscle characteristics is still debated: according to a recent study, repeated BoNT-A injection cycles did not seem to induce fibroadipose infiltration, and the muscle degeneration documented was more likely related to spastic muscle evolution [37]. Alternatively, muscle biopsies of BoNTA-treated muscles in humans with underlying medical conditions have shown variable outcomes [38].

Hence, when implementing long-term treatment for spasticity, it should be considered that the characteristics of the inoculated muscle substrate could change, either due to the evolution of the pathology and/or as an undesirable effect of the BoNT-A treatment. Consequently, the response to treatment may also change over time.

Based on these premises, we believe that in clinical practice, it would be appropriate to monitor the evolution of muscle changes over time and accordingly guide the choice of treatment.

Undoubtedly, a complete and in-depth study of the muscle would require the use of invasive (e.g., muscle biopsy) or expensive (CT, MRI, PET) assessment modalities. These tests would provide a lot of useful information about muscle composition and macroscopic muscle changes, but they are not easily applicable to a clinical routine.

A safe, non-invasive, and less expensive alternative is musculoskeletal ultrasound, which has been shown to be a reliable method to determine the severity of structural muscular changes thanks to its sensitivity to fibrous tissue [39].

Picelli et al. found that patients with a higher spastic muscle echo intensity may have a reduced response to BoNT-A [40], which suggests that, in the context of the long-term treatment of PSS with BoNT-A, the routine use of this method could allow the physician to identify which patients would benefit most from the treatment (albeit by increasing the dosage), as well as those for whom alternative therapeutic interventions should be considered.

Finally, we observed that the mean time between stroke onset and the beginning of BoNT-A treatment amounts to 3.4 ± 3.5 years.

It is well known in the literature that spasticity can occur within the first weeks after a stroke and that it can result in pain, contractures, and bedsores, further compromising the patient’s quality of life [41].

In this regard, many studies underline the importance of early diagnosis and treatment since the following is true:(1)Many post-stroke complications are either preventable or may be potentially ameliorated with treatment [42,43,44,45];(2)The early management of PSS may also improve function and increase independence in post-stroke patients [46].

Although we could not provide a formal analysis supporting this view, in our patient cohort, PSS was probably often detected and treated too late when, presumably, incorrect movement patterns and abnormal limb posture had already been established.

This highlights an important problem: most likely, in local settings, there is little awareness of spasticity predictors and the impact that an early diagnosis and treatment can have on a patient’s clinical evolution. Furthermore, the gradual onset of spasticity could lead less experienced physicians to underestimate its first signs and delay the start of treatment, indicating the need for increased dissemination of knowledge about this issue. Finally, it could be that, in Italy, the number of specialized spasticity services is very small compared to requirements, which could contribute to further delays in starting treatment due to difficulties in accessing these services.

Therefore, raising the awareness of physicians operating in local settings on this topic and encouraging the establishment of an efficient network of centers specialized in the treatment of PSS with BoNT-A must represent future objectives. 

We are aware that this study has several limitations.

Firstly, due to its retrospective design and reliance on medical chart reviews, we were only able to analyze certain aspects of long-term treatment with Botulinum Toxin type A. Important medical information, such as comorbidities and concurrent treatments (e.g., systemic muscle relaxant, nerve blocks with phenol or alcohol), were not consistently available. The concomitant use of muscle relaxant drugs contributes to a reduction in spasticity and may prolong the effectiveness of Botulinum Toxin, influencing the clinical outcome. The limited available data relating to muscle relaxant drugs were not sufficient to draw firm conclusions; however, as already mentioned in the results, they allowed us to minimize any bias arising from the assumption of using or not using these drugs. Additionally, a further limitation of our study is that we had no data available regarding the impact of long-term treatment on functional recovery and any structural changes in inoculated muscles. The absence of these variable factors prevented us from drawing exhaustive conclusions about the long-term effects of BoNT-A and may introduce bias in the medication intervals and dosage outcomes.

Nevertheless, this is not in conflict with our study’s mission, which aims to describe the overall trends in spasticity management to date.

Further research is needed to assess the impact of long-term BoNT-A treatment on the management of PSS and on functional recovery. Moreover, a prospective longitudinal design with follow-up ultrasonographic assessments could allow a better understanding of the causes underlying the reduction/loss in efficacy of BoNT-A and could push toward more customized treatment.

## 4. Conclusions

In conclusion, our study offers significant implications for the management of long-acting spasticity treated with BoNT-A. In particular, the following was found:(1)Long-term BoNT-A treatment modulates dosages;(2)Most dropouts due to effectiveness occur in the first years of treatment;(3)Early intervention is crucial, as well as a clear definition of the objectives of the treatment with the patient, to increase their adherence to the therapy;(4)Since muscular structure can change over time, a periodic evaluation of BoNT-A treatment response is needed.

## 5. Materials and Methods

We performed a retrospective medical records review of patients who were treated for PSS in our clinic from November 2006 to February 2024. Inclusion criteria were (I) ischemic or hemorrhagic stroke in a chronic phase; (II) upper and/or lower limb spasticity (MAS comprised between 1+ and 3); (III) treatment with BoNT-A repetitive injections for at least 2 years (long-term patients); (IV) treatment with the same formulation of BoNT-A for the whole observation period; and (V) an ability to provide informed consent.

Data from 150 medical records were collected; we excluded 45 patients who were not considered “long-term patients” (<2 years of treatment) and, in order to avoid any bias due to the different potency of the three formulations, we excluded 10 patients who had been treated with two or more types of BoNT-A. Therefore, 95 patients were included in our analysis (Figure 4). We collected information based on age, gender, disease duration, treatment duration, and the types of BoNT-A injected. We also measured the time interval between consecutive BoNT-A injections (in months) and then calculated, for each year of treatment, the average of all the intervals that emerged.

Finally, we collected data on the total dosage administered during each session (Figure 3) and calculated the average for each year of treatment (Figure A2 in Appendix B). Before proceeding with BoNT-A injections, each patient signed an informed consent form and underwent a complete clinical evaluation, including the evaluation of spasticity by means of the Modified Ashworth Scale (MAS).

Toxin injections were performed by a well-trained physiatrist who had previously identified the optimal dose and the target muscles according to the clinical evaluation of spasticity.

OnabotulinumtoxinA (BOTOX^®^), AbobotulinumtoxinA (DYSPORT^®^), or IncobotulinumtoxinA (XEOMIN^®^) were used to treat the patients in our cohort.

Within the limits of a retrospective study, there can be bias due to the inclusion of patients treated with different formulations of toxin; however, in our case, this factor should not significantly influence the results as the types of toxin were fairly evenly distributed across the entire group and within the two subgroups (see Appendix A).

Patients were divided into two different groups based on the duration of treatment. The first group comprised patients with a duration of treatment greater than 2 and less than 5 years (Short Interval Treatment group, SIT, n = 48); the second group comprised patients with a duration of treatment equal to or greater than 5 years (Long Interval Treatment group, LIT, n = 47). The two groups differed considerably in their time range of treatment (SIT: 2–4 years; LIT: 5–14 years), but this choice was necessary to create groups of a similar sample size. It is also important to remember that both groups were composed of patients involved in long-term treatment. This stratification allowed us to investigate the effect of the duration of treatment on two dependent variables, namely, the interval between injections (in months) and the dosage of each injection (UI). We also investigated the temporal evolution of the dropouts, as well as their reasons.

We used the Shapiro–Wilk test to assess the normality of the data distributions. Most distributions violated the assumption of normality. Hence, we used non-parametric tests, namely, the Wilcoxon signed-rank test, to compare the effects of treatment over time within a group and the Mann–Whitney test for the comparison of treatment effects between groups. Whenever required, we corrected for multiple comparisons using Benjamini/Hochberg FDR correction [47].

Data were analyzed using R Statistical Software [48] (v 4.2.3). We set statistical significance at *p* < 0.05. Unless otherwise stated, the results in the following sections are reported as the median (M) and interquartile range (IQR), i.e., M{IQR}, where
IQR = Q75 − Q25(1)
and Q75 and Q25 are the 75th and 25th percentiles of the data distribution, respectively.

## Figures and Tables

**Figure 1 toxins-16-00383-f001:**
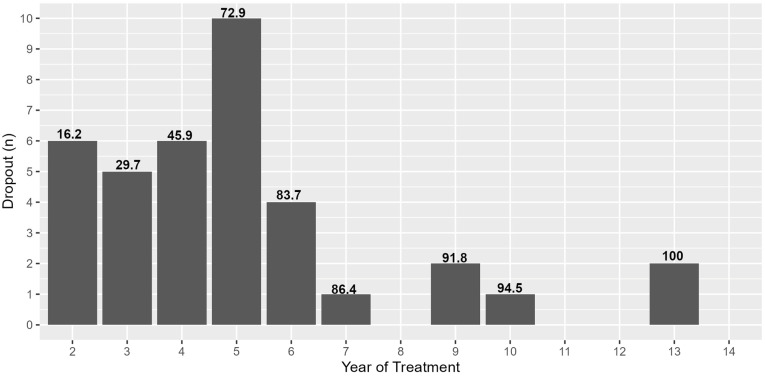
Number of dropouts by year of treatment. Numbers above the bars indicate the cumulative percentage of dropouts.

**Figure 2 toxins-16-00383-f002:**
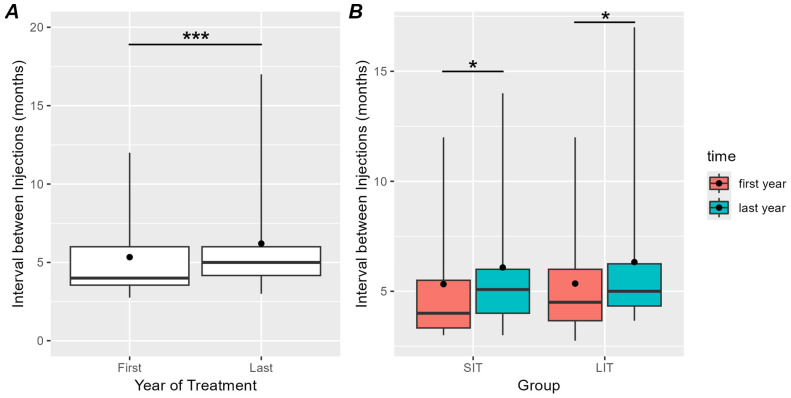
(**A**) Distribution of the mean intervals between injections in the first and last year of treatment in all patients. Horizontal lines indicate medians and black dots indicate means. Whiskers extend to points that lie within 1.5 interquartile ranges of the lower and upper quartiles. (**B**) Distribution of the mean intervals between injections in the first (red) and last year (cyan) of treatment separately for the SIT and LIT groups. The same graphical conventions are used as in panel (**A**). Asterisks indicate larger intervals in the last year of treatment than in the first. ***, *p* < 0.001, *, *p* < 0.05.

**Figure 3 toxins-16-00383-f003:**
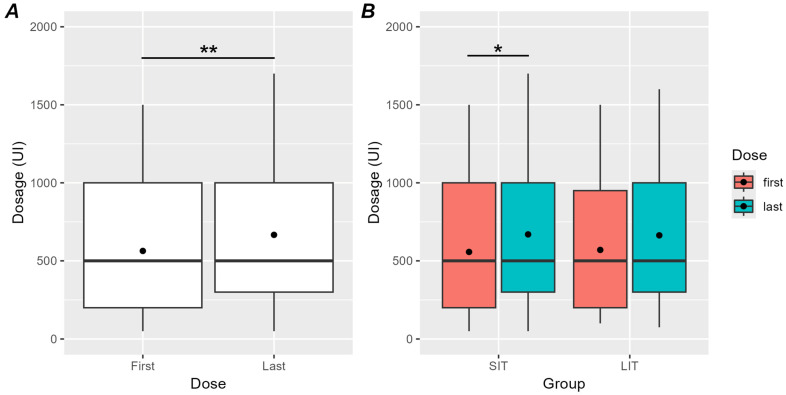
(**A**) Distribution of the dosage in the first and in the last injection in all patients. Horizontal lines indicate medians and black dots indicate means. Whiskers extend to points that lie within 1.5 interquartile ranges of the lower and upper quartiles. (**B**) Distribution of the dosage in the first (red) and last injection (cyan) of treatment separately for the SIT and LIT groups. The same graphical conventions were are as in panel (**A**). Asterisks indicate a greater dosage in the last injection than in the first. **, *p* < 0.01, *, *p* < 0.05.

**Figure 4 toxins-16-00383-f004:**
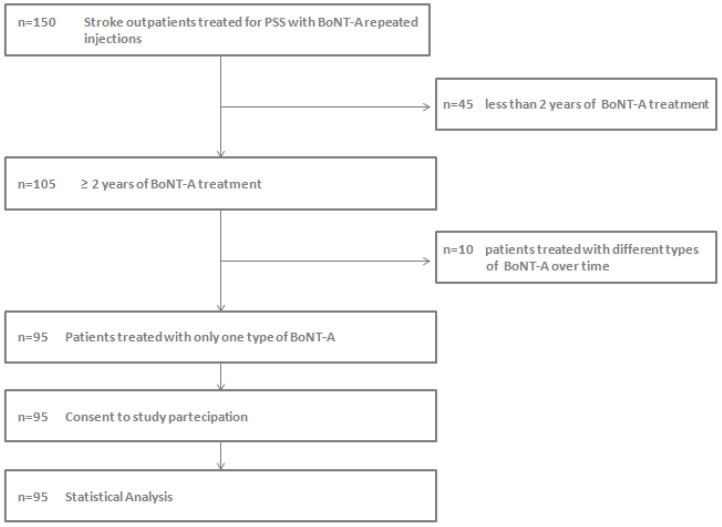
Flow chart of the retrospective patients’ selection for data acquisition and statistical analysis.

**Table 1 toxins-16-00383-t001:** Demographics and treatment-related data (n = 95).

Gender, n (%)	
Female	37 (38.9)
Male	58 (61.1)
Age (years)	63.2 ± 12.5
mean ± SD, range	22–85
Duration of illness	9.2 ± 5.3
mean ± SD	
Duration of treatment (years)	5.3 ± 3.1
mean ± SD, range	2–14
Mean time between stroke onset and first treatment (years)	3.4 ± 3.5
mean ± SD	

**Table 2 toxins-16-00383-t002:** Dropout.

Group ^1^	SIT	LIT
Dropout, n, % ^2^	17	20
	46	54
Dropout reason ^3^, n	U = 12	U = 12
	I = 4	I = 5
	D = 1	D = 3

^1^ SIT = Short Interval Treatment group; LIT = Long Interval Treatment group. ^2^ % dropout percentage relative to the whole sample of participants who dropped out (n = 37). ^3^ U = unknown; I = ineffective treatment; and D = death.

## Data Availability

The raw data supporting the conclusions of this article will be made available by the authors on request.

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
