# Peer review of "Long-Term Management of Post-Stroke Spasticity with Botulinum Toxin: A Retrospective Study"

_toxins, 2024, doi:10.3390/toxins16090383_

Round 1

Reviewer 1 Report (Previous Reviewer 2)

Comments and Suggestions for Authors

1. Although the statistics has been greatly improved in the revised paper, the lack variables related to anti-spasticity treatment in the design lead to the lower reliability of the conclusions.

2.This retrospective study focused on the temporal evolution of treatment intervals, dosage patterns, and dropout rates of Botulinum Toxin. As we knew, there are commonly clinical anti-spasticity treatment combined, including anti-spasticity drug, assistive devices ,robots and so on,which may lead to prolonged Botulinum Toxin treatment duration.

3. It is concluded that the dosage of Botulinum Toxin increased with the prolongation of the course of disease and the aggravation of muscle spasm, which seemed to be inconsistent with the conclusion that the treatment duration prolonged.

4. In the study, early Intervention is suggested for post-stroke spasticity, but there does not appear to be a supporting data analysis for this view.

5. In the discussion part, the author suggested that musculoskeletal ultrasound would be useful in determining whether the benefit of Botulinum Toxin therapy should be continued. However, there is also no relevant data in the study for this view.

The suggestions are available for the author. Please add the important data of treatment in muscle spasm, and we suggest that the outcome analysis consistent with the data.

Author Response

Reviewer 2 Report (New Reviewer)

Comments and Suggestions for Authors

1. Provide more comprehensive details on the criteria for determining treatment intervals and dosages, and justify patient exclusions more thoroughly.

2. Please include a comparative analysis between BoNT-A therapy and the other treatment options for spasticity that you mention in the introduction, such as muscle relaxants, ESWT, non-invasive brain stimulation, and localized muscle vibration. Such a discussion would offer readers a clearer understanding of BoNT-A's unique advantages and limitations compared to these alternative therapies.

            For example, see the following paper: Mihai EE, Popescu MN, Iliescu AN, Berteanu M. A systematic review on extracorporeal shock wave therapy and botulinum toxin for spasticity treatment: a comparison on efficacy. Eur J Phys Rehabil Med. 2022 Aug;58(4):565-574. doi: 10.23736/S1973-9087.22.07136-2. Epub 2022 Apr 12. PMID: 35412036; PMCID: PMC9980509.

Comments on the Quality of English Language

The manuscript would benefit from a more cohesive and fluid writing style. Currently, the use of very short sentences disrupts the flow of the text, making it difficult to read and follow the argumentation. To enhance readability and improve the overall quality of the manuscript, I recommend revising the sentence structure to create smoother transitions between ideas. This might involve combining shorter sentences into more complex ones where appropriate, and ensuring that each paragraph develops a clear and connected thought.

Additionally, I strongly suggest that the manuscript be reviewed by a native English speaker or a professional editor with experience in academic writing. This would help to address any grammatical issues and ensure that the language is clear, precise, and suitable for publication in a scientific journal.

Author Response

Reviewer 3 Report (New Reviewer)

Comments and Suggestions for Authors

In summary, the authors  studied the long term evolution (effect and management) of 95 patients with stroke-related spasticity treated with botulinum toxin A (TBA). As is the general norm, This is retrospective  study analyzing medical records medical records of 95 chronic stroke patients undergoing long-term BoNT-A

The study revealed several interesting findings such as  the notable drop-out rate across the time and  trend of escalating dosage probably related to rise in spasticity severity or changes in treatment response over time. The authors comment on these findings and suggest that botulinum toxin treatment begins to late (average 3 years after stroke) and that the patients expectations and goals probably must be clarified from the start. They also comment the dynamic nature of spasticity

I suggest several points:

1. As the authors comment, spasticity is not a static process, just like dystonia, spasticity evolves over time, the muscle changes and fibrosis increases the rigidity  and mechanical characteristics, hence, the action of BTA probably decreases over time….this may partly explain the increasing dose and drop-out rate .

2. Spasticity is beneficial or not ?  Interesting question raised by the authors…in general terms, the main late problem is not spasticity but paresis…I suggest to discuss the fact that spasticity may resolve …or increase over time

3. The objectives and expectations of the patients are critical issues already raised by the authors. Certainly the first contact is crucial  for the patient and caregivers

4. Minor advice: Do not duplicate information in the text and tables

5. Several concise, clear bullets would be important as take-away messages

Such as:

a)  If BTA is needed…  start as soon as possible

b)  Periodic evaluation is needed. What is the actual problem ?  spasticity or paresis…Does my patient still need the treatment ?

c)  Objective clear and concise from the start

In any case, an interesting paper that could improve  if several items are further discussed

Round 2

Reviewer 1 Report (Previous Reviewer 2)

Comments and Suggestions for Authors

  Compared with the previous manuscript,the author made a supplement of data about the anti-spastic drugs (Baclofen, Tizanidine or Eperisone Hydrochloride) during the treatment with BoNT-A in the sample analysis.And in the discussion part, authors focused on observing potential changes in BoNT-A treatment over time, in terms of dosage variability and administration interval.

   In this retrospective study, the authors mentioned that a notable extension of the interval between BoNT-A injections throughout the treatment duration. Dropout rates peaked during the initial 5 years of treatment.And a trend of escalating dosage was observed across all groups.Then the authors addressed some suggestions,including the importance of an early intervention and periodic evaluation of BoNT-A treatment response is needed due to muscular structure change over long time.These clinical meanings could be helpful for the medical health workers.We intend to accept this manuscript to be published.

Reviewer 2 Report (New Reviewer)

Comments and Suggestions for Authors

I have carefully reviewed the authors' revisions and their responses to the referees' comments. I am pleased to inform you that the manuscript has been sufficiently improved and is now suitable for publication. Congratulations! 

This manuscript is a resubmission of an earlier submission. The following is a list of the peer review reports and author responses from that submission.

Round 1

Reviewer 1 Report

Comments and Suggestions for Authors

The authors are to be commended on an excellent review.

Reviewer 2 Report

Comments and Suggestions for Authors

This study were designed to explore the role of Botulinum Toxin injections in long-term management of spasticity in chronic post-stroke patients. The author performed a review of 95 patients with chronic stroke, who were treated with repeated injections of BoNT-A for a minimum of 2 years and a maximum of 14 years. However, the overall research design was not rigorous, the important medical information of patients were incomplete, and important variables were not considered, including lack of patient flowchart, comorbidities, other anticonvulsant drugs such as Baclofen, Tizanidine or other therapies, nerve block with phenol or alcohol, treatment with steroids or aminoglycosides. And the lacks of those variable factors would probably induce the over- or underestimation of the effect, and a bias against the results of botulinum toxin medication interval and dosage. Therefore, we consider the main conclusions lack important evidence, and recommend a rejection.

Comments on the Quality of English Language

None

Reviewer 3 Report

Comments and Suggestions for Authors

lines 36 - 58: in the introduction, three paragraphs are devoted to a detailed description of spasticity, in my opinion, it is not closely related to the topic of the work and providing detailed information is unnecessary in this case. It would be good to include information on the differences in the toxic power of various preparations containing BoNT/A, which may influence the design of the experiment

As the authors say, this is not the only manuscript describing the long-term effect of using botulinum toxins for over two years. Please verify the literature data, once again.

Material and Methods

The materials and methods should include information on standardization between different botulinum toxin preparations. Three preparations of botulinum toxin were used, they may have other effects due to different excipients and toxic power. There is also no information on individual preparations in a given number of patients.
line 33 - 344 - division into SIT, MIT and LIT groups is made based on arbitrarily assigned treatment periods. The groups are not equal or close in size. Therefore, the statistical comparison of the mentioned groups is not entirely reliable. Wouldn't it be better to consider retrospectively periods with equal or similar numbers, e.g. combining MIT and LIT.

Results

Please, check the results once again. There are mistakes and discrepancies between lines 108-112 and Table 2.

Discussion and conclusions

The discussion and conclusions lack a specific effect resulting from the research conducted by the authors. The mere observation of an increase in the number of injection intervals in a time frame is not a significant and revealing conclusion.

References

Please check literature data about long - term use of botulinum toxin and toxic power and differences between BoNT/A preparations.

Literature formatting is incorrect